# LAM Cells as Potential Drivers of Senescence in Lymphangioleiomyomatosis Microenvironment

**DOI:** 10.3390/ijms23137040

**Published:** 2022-06-24

**Authors:** Clara Bernardelli, Silvia Ancona, Melania Lazzari, Antonella Lettieri, Piera Selvaggio, Valentina Massa, Cristina Gervasini, Fabiano Di Marco, Raffaella Chiaramonte, Elena Lesma

**Affiliations:** 1Department of Health Sciences, Università degli Studi di Milano, 20142 Milan, Italy; clara.bernardelli@unimi.it (C.B.); silvia.ancona@unimi.it (S.A.); melania.lazzari96@gmail.com (M.L.); antonella.lettieri@unimi.it (A.L.); pieraselvaggio@gmail.com (P.S.); valentina.massa@unimi.it (V.M.); cristina.gervasini@unimi.it (C.G.); fabiano.dimarco@unimi.it (F.D.M.); raffaella.chiaramonte@unimi.it (R.C.); 2“Aldo Ravelli” Center for Neurotechnology and Experimental Brain Therapeutics, Università degli Studi di Milano, 20142 Milan, Italy; 3Respiratory Unit, Azienda Socio Sanitaria Territoriale-Papa Giovanni XXIII Hospital, 24127 Bergamo, Italy

**Keywords:** senescence, LAM, mTOR, tuberin, SASP

## Abstract

Senescence is a stress-response process characterized by the irreversible inhibition of cell proliferation, associated to the acquisition of a senescence-associated secretory phenotype (SASP), that may drive pathological conditions. Lymphangioleiomyomatosis (LAM) is a rare disease in which LAM cells, featuring the hyperactivation of the mammalian Target of Rapamycin Complex 1 (mTORC1) for the absence of tuberin expression, cause the disruption of the lung parenchyma. Considering that LAM cells secrete SASP factors and that mTOR is also a driver of senescence, we deepened the contribution of senescence in LAM cell phenotype. We firstly demonstrated that human primary tuberin-deficient LAM cells (LAM/TSC cells) have senescent features depending on mTOR hyperactivation, since their high positivity to SA-β galactosidase and to phospho-histone H2A.X are reduced by inducing tuberin expression and by inhibiting mTOR with rapamycin. Then, we demonstrated the capability of LAM/TSC cells to induce senescence. Indeed, primary lung fibroblasts (PLFs) grown in LAM/TSC conditioned medium increased the positivity to SA-β galactosidase and to phospho-histone H2A.X, as well as p21^WAF1/CIP1^ expression, and enhanced the mRNA expression and the secretion of the SASP component IL-8. Taken together, these data make senescence a novel field of study to understand LAM development and progression.

## 1. Introduction

Cellular senescence is a physiological stress response in which cells are in a permanent status of cycle arrest. This process is highly dynamic, and it represents an important mechanism through which tissue homeostasis is maintained, as it regulates tissue remodelling and repair during development and in adulthood [1]. Nevertheless, an irreversible proliferation arrest might also contribute to the loss of tissue homeostasis by limiting its regenerative capacity, inducing a pro-inflammatory status that might sustain the onset of pathological condition and age-related diseases, i.e., atherosclerosis and pulmonary idiopathic fibrosis (IPF) [2]. Besides as the inhibition of cell cycle, senescence should be considered also as the activation of protein secretion and cell flattening. Senescent cells acquire the capability to alter the functions of neighbouring cells through the release of senescence-associated secretory phenotype (SASP) factors, composed of pro-inflammatory cytokines, chemokines, growth factors, and matrix metalloproteinases. These molecules promote inflammation, invasion, angiogenesis, and cell proliferation, which might drive the development of several pathologies, including cancer [3]. Indeed, senescent fibroblasts induced by genotoxic stimuli, such as chemotherapy, might sustain the proliferation of cancer cells and enhance their invasive capability through the secretion of specific SASP components [4]. Among those, the pro-inflammatory cytokine interleukin 8 (IL-8) is shown to act as an autocrine agent, which reinforces the senescent phenotype of IMR90 fibroblasts through the signalling of its receptor CXCR2 [5]. This evidence makes this protein an important mediator for the spreading of the SASP response in the microenvironment [6]. At the same time, the presence of pro-inflammatory molecules in the SASP (such as the above-mentioned IL-8) can induce a chronic inflammatory state, and it attracts immune cells that disrupt the local microenvironment [2]. In this case, the loss of proliferative potential of senescent cells impairs tissue regeneration and promotes aging.

Lymphangioleiomyomatosis (LAM) is a multisystem rare disease in which smooth muscle-like LAM cells invade lungs, causing the disruption of the lung parenchyma and leading to respiratory failure [7]. Renal angiomyolipomas and an involvement of the lymphatic system with the onset of lymphangioleiomyomas, chylous effusions, and adenopathy may also occur [8]. In LAM cells, the loss of heterozygosity (LOH) in the tuberous sclerosis complex (*TSC*) *1* or *TSC2* tumour suppressor genes, which code for hamartin and tuberin, respectively, leads to the hyperactivation of mammalian Target of Rapamycin Complex1 (mTORC1) for the absence of one of the two proteins [7]. mTOR signalling is crucial for cell growth and proliferation since it induces cellular responses to stress, including the regulation of the metabolic activity and the contribution to the senescence establishment [9]. Recent studies demonstrated that LAM cells destroy the lung tissue by proteolytic activity through the secretion of cathepsin K [10] and metalloproteinases [11,12], whose secretion is also constitutively upregulated in SASP [3]. Similarly, the Vascular Endothelial Growth Factor-D (VEGF-D), a known biomarker for LAM [13], is a key component of SASP, being highly secreted from senescent fibroblasts to promote wound healing in human skin [14]. Moreover, IL-6 is the most represented cytokine in the SASP [15] and it is highly secreted by LAM/TSC cells, as well as IL-8 [16]. Recently, LAM has been reclassified as a low-grade, destructive, metastasizing neoplasm, considering that LAM and cancer share genetic and cellular features, such as the clonal origin of the cells, the metabolic reprogramming, the invasive and migratory capability, and the relapse [17]. However, LAM differs from other neoplasms for the bilateral and symmetrical disruption of the lung without a clear primary tumour site and dominant mass lesions. On these bases, senescence, and in particular the SASP, might have a role in driving disease progression. Indeed, even if it is demonstrated that LAM cells enhance the capability of lung fibroblasts to promote lung epithelial cell migration and proliferation [18], the mechanisms underlying the pathological communication are still not completely understood.

The aim of this study is to deepen the possible role of senescence in LAM pathogenesis. We analysed some specific senescent-associated hallmarks in primary tuberin-deficient LAM cells derived from the chylothorax of a LAM patient (LAM/TSC cells) [16] and in human Primary Lung Fibroblasts (PLFs), obtained from a lung biopsy, that express tuberin. We demonstrated that LAM cells have senescent features, as shown by the SA-β-galactosidase activity, the high percentage of cells positive to phosphorylated histone H2A.X, and the high IL-8 expression and secretion. Since in LAM/TSC cells the absence of tuberin is caused by an epigenetic second hit due to the hypermethylation of the *TSC2* promoter, the chromatin-remodelling agent 5-azacytidine was used to induce tuberin expression, causing a significant reduction of senescent features, similarly to the inhibition of mTOR by rapamycin. LAM/TSC cells can also induce senescence on PLFs. Indeed, by growing PLFs in the Conditioned Medium (CM) of LAM/TSC cells, proliferation was significantly reduced, and this decrease was related to significantly higher activation of SA-β-galactosidase, higher histone H2A.X phosphorylation, and higher levels of the senescent marker p21^WAF1/CIP1^ expression. Finally, PLFs grown in LAM/TSC cell CM are induced to express and secrete high amount of the pro-inflammatory and senescence-inducing-SASP component IL-8.

The demonstration of senescent features in LAM cells and of their capability to propagate senescence phenotype in fibroblasts introduces an interesting approach to investigate novel mechanisms to modulate LAM progression.

## 2. Results

### 2.1. Evaluation of Senescence in LAM/TSC Cells and PLFs

Senescence is a heterogenous process that needs to be analysed by using multiple approaches since no single universal marker has been identified. The expression of SA-β-galactosidase is a good biomarker to detect senescent cells, both in culture and in tissues, with a histochemical assay [19]. Moreover, the phosphorylation of the histone H2A.X on Ser139 occurs in senescent cells to promote the assembly of DNA repair and checkpoint factors, thus initiating the senescence program in response to DNA damage [20]. To investigate if senescence could contribute to the LAM development and progression, we studied the senescent features of a LAM cell line, the human primary LAM/TSC cells isolated from chylous thorax of a patient affected by LAM associated to TSC [16], and of PLFs, obtained from a lung biopsy, which express tuberin and do not display mTOR hyperactivation (Figure 1a). Comparing these LAM/TSC cells to PLFs, we observed a significantly higher level of SA-β-galactosidase staining (Figure 1b) and a higher percentage of phospho-histone H2A.X-positive cells compared with PLFs (Figure 1c). These data sustain a LAM/TSC cell senescent behaviour, and make these cells a good model to study senescence in LAM.

### 2.2. Effect of the Induction of Tuberin Expression and mTOR Inhibition on LAM/TSC Senescent Features

The lack of tuberin leads to the hyperactivation of mTOR, which exerts many functions in controlling cell growth and metabolism, and that has a role in sustaining senescence, being considered a positive regulator of SASP [21]. LAM/TSC cells do not express tuberin because of a germline *TSC2* mutation and the hypermethylation of the *TSC2* promoter, making these cells an interesting model in which restore tuberin expression by the exposure to chromatin remodelling agents such as 5-azacitydine [16]. To analyse if the LAM/TSC cell senescent features might depend on the mTOR hyperactivation, senescence was firstly evaluated following the induction of tuberin expression. As expected, the phosphorylation of S6, the substrate of S6 kinase (S6K), a functional marker of mTOR activity [22], was reduced by 1 µM 5-azacytidine treatment for 96 h, which caused tuberin expression (Figure 2a). Interestingly, also the levels of the phosphorylated form of ZRF1, which is a substrate of S6K that drives senescence reprogramming in fibroblasts [23], were impaired, while p21^WAF1/CIP1^ expression was not significantly affected (Figure 2b), likely suggesting that tuberin expression only influences the pathways downstream mTOR but it does not have effects on the cell cycle arrest in senescence. As further confirmation of this hypothesis, a significant decrease of the positivity to the SA-β-galactosidase staining was observed in tuberin-expressing LAM/TSC cells following 5-azacytidine treatment (Figure 2c), while the induction of tuberin only slightly reduced the cells positive to phospho-histone H2A.X (5%) compared to tuberin null LAM/TSC cells (7.8%) analysed by flow cytometry (data not shown). Of note, the treatment with 5-azacytidine did not significantly affect LAM/TSC cell viability, as demonstrated by trypan blue exclusion assay, in accordance with our previous study [16]. Moreover, the incubation with 5-azacytidine for 96 h did not cause a relevant apoptotic process as demonstrated by the absence of cleaved fragments of the apoptotic markers Caspase 3 and PARP by Western Blot and by the low positivity to annexin V analysed by flow cytometry (Appendix A).

To confirm that the expression of tuberin can decrease the senescent features of LAM/TSC cells through the modulation of mTOR activity, LAM/TSC cells were grown for 48 h with the mTOR inhibitor rapamycin at the concentration of 0.5 ng/mL and 1 ng/mL. Similar to the data shown above, rapamycin caused the reduction of S6 and ZRF1 phosphorylation, and inhibited the p21^WAF1/CIP1^ expression even if only at the higher drug concentration (Figure 3a,b). Consistently, the percentage of SA-β-galactosidase positive LAM/TSC cells was reduced by rapamycin treatment (Figure 3c) as well as the percentage of cells expressing the phosphorylated histone H2A.X analysed by flow cytometry (3.04% and 1.95% for LAM/TSC cells treated with 0.5 ng/mL and 1 ng/mL of rapamycin, respectively, and 7.8% in control LAM/TSC cells) (data not shown). LAM/TSC cell viability, evaluated by trypan blue exclusion assay, was not significantly affected by the 48-h incubation with 0.5 ng/mL or 1 ng/mL of rapamycin (Appendix A), consistently with our previous demonstration that, at this time point, the proliferation rate of LAM/TSC cells in the presence of rapamycin concentrations up to 5 ng/mL is similar to controls [16]. As shown for 5-azacytidine treatment, the 48-h treatment with 0.5 ng/mL or 1 ng/mL of rapamycin did not induce apoptosis in LAM/TSC cells (Appendix A).

### 2.3. Evaluation of the Effect of Conditioned Medium (CM) of LAM/TSC Cells on PLFs Proliferation and Apoptosis

Considering that LAM cells represent only a small population in the lung LAM microenvironment but cause a diffuse parenchyma disruption [7], we developed an in vitro model to investigate the capability of LAM/TSC cells to influence PLFs, by likely inducing a LAM-supporting microenvironment. PLFs were grown in the CM of LAM/TSC cells to mimic a paracrine communication between these two cellular populations. As control, PLFs were grown on their own CM, in order to avoid artefacts due to the culture in an exhausted medium. Firstly, LAM/TSC cell CM reduced the PLFs proliferation, with a significant decrease in PLFs number starting from 48 h (Figure 4a,b). The delayed PLFs growth in LAM/TSC cell CM was not due to an increased PLFs death, as demonstrated by the unaltered levels of full-length apoptotic markers Caspase 3 and PARP, by any significant difference in their cleaved fragments (Figure 4c) and the low positivity to annexin V, which is not significantly different between PLFs grown on their own CM and in LAM/TSC cell CM (Figure 4d).

### 2.4. Effect of LAM/TSC Cell CM on Senescent Features of PLFs

To investigate if the delay of PLFs proliferation might be due to the induction of a senescent phenotype caused by LAM/TSC cell CM, senescent markers were studied. The positivity to SA-β-galactosidase and to phospho-histone H2A.X of PLFs grown for 48 h in LAM/TSC cell CM was significantly higher compared to controls (Figure 5a,b). To confirm the capability of LAM/TSC cell CM to induce senescence, IMR90 cells, a fibroblast cell line usually used as a senescence model [5], were grown for 48 h in the CM from LAM/TSC cells. Similar to what was observed for PLFs, the IMR90 cells positive to SA-β-galactosidase and to phospho-histone H2A.X were significantly increased (Figure 5c,d). It should be considered that the SA-β-galactosidase activity can be influenced by high levels of glucose (e.g., 22 mM) and in general by medium components [24]. Since IMR90 cells grow in DMEM and LAM/TSC cell CM was composed of Type II medium, senescent features of IMR90 grown in their own Type II CM and in their own DMEM CM were compared to evaluate the influence of the medium composition. We did not observe any significant differences between the two groups, indicating that the different DMEM and Type II medium formulations, at our experimental conditions, did not influence the modulation of senescence as evaluated by the used hallmarks (Figure 5c,d: columns 1 and 2). Likely, the difference between the glucose concentrations in the two media is not high enough to influence senescence. DMEM contains 1 g/L (5 mM) of glucose, lower than Type II medium that is composed by 1:1 mixture of DMEM and Ham’s F12, whose glucose concentration is 1.802 g/L (10 mM). Moreover, the different concentration of the serum, which has the same source and is 10% in DMEM and 15% in Type II medium, appears to not be relevant in our experimental conditions to modulate senescence.

To better study the effect of LAM/TSC cell CM on senescence in this model, p21^WAF1/CIP1^ and p16^INK4A^ expression, biomarkers involved in signalling pathway specific for cell-cycle arrest, which inhibit cyclin-dependent kinase, were evaluated by Western Blot. The induction of senescence might be driven by p21^WAF1/CIP1^, whose expression is increased upon PLFs cultured in LAM/TSC cell CM, while, surprisingly, p16^INK4A^ was unaltered (Figure 6a,b). mTOR activity in PLFs seems to be unaffected by 48 h of growing in LAM/TSC CM, since S6 phosphorylation was not altered (Figure 6c).

### 2.5. Evaluation of IL-8 Expression and Secretion in PLFs, LAM/TSC Cells, and in PLFs Grown in CM

IL-8 is a chemokine overexpressed in the most senescent cells as a SASP component and its high levels in lung parenchyma have been related to the pathogenesis and progression of lung diseases, such as IPF [25]. IL-8 is expressed in various cell types, including fibroblasts, and is highly secreted by LAM/TSC cells [16,26]. The mRNA relative expression and the secretion of IL-8 were significantly higher in LAM/TSC cells compared to PLFs suggesting a SASP IL-8 role in sustaining LAM/TSC cell senescent phenotype (Figure 7a,c). When PLFs were grown in LAM/TSC cell CM, the IL-8 mRNA relative expression significantly increased (Figure 7b). In a similar way, PLFs grown in LAM/TSC cell CM secreted higher IL-8 levels than PLFs grown in their CM (Figure 7c). Considering that the CMs contain IL-8 secreted in 72 h, in this latter case, IL-8 levels of PLFs or of LAM/TSC cells were subtracted to IL-8 levels of PLFs grown for 48 h in their CM or in LAM/TSC cell CM.

## 3. Discussion

In this study, we demonstrated that human primary tuberin-deficient LAM/TSC cells have a senescent phenotype, as shown by the activation of the SA-β-galactosidase enzyme and by the phosphorylation of the histone H2A.X, which are both significantly higher in LAM/TSC cells compared to human PLFs, which do not have hyperactivation of mTOR. The inhibitory modulation of the senescent features by rapamycin and by the induction of tuberin expression through 5-azacytdine indicated that, in LAM/TSC cells, senescence is dependent on mTOR hyperactivation. Finally, we demonstrated that LAM/TSC cell CM can induce senescence in PLFs, with a key role of the SASP factor IL-8, which is highly expressed and secreted in LAM/TSC cells and PLFs grown in LAM/TSC cell CM.

Among the physiological and pathological processes involved in the LAM onset, we focused on senescence, hypothesizing that it might promote the loss of the parenchymal structure in the lungs, causing a severe impairment of the lung function, as already demonstrated for other respiratory diseases, such as IPF [2]. Senescence could be a novel mechanism to explain the highly variable manifestation and progression of LAM. At the state of the art, this hypothesis was only proposed but not demonstrated so far [27]. Indeed, it is already known that LAM cells exploit several mechanisms to induce the remodelling of the lung parenchyma, including the secretion of molecules such as matrix metalloproteinases [11], kathepsin K [10], VEGF-D [7], IL-6 [28], and IL-8 [16], which are also SASP components released by senescent cells to modulate the behaviour of neighbouring cells.

Moreover, as further confirmation of this hypothesis, the evidence of an early senescent phenotype in mouse Tsc2-null embryo fibroblasts (MEF lines), which bear a constitutive activation of mTORC1 due to loss-of-function mutations in *TSC1*, *TSC2,* or *PTEN,* was reported [29]. The absence of tuberin in LAM/TSC cells leads to mTOR hyperactivation likely causing senescence. Indeed, senescent cells might overcome the cell cycle arrest through the hyperactivation of the growth-promoting mTOR pathway [30], leading to lysosomal hyperfunction that can be detected following the activity of the lysosomal protein SA-β-galactosidase, used as senescence biomarker in senescent fibroblasts and keratinocytes, as well as in aged skin in vivo [31]. In LAM/TSC cells, the hyperactivation of mTOR appears to play a central role in senescence, as demonstrated by the decreased SA-β galactosidase positivity after the downregulation of mTOR by inducing tuberin expression or by using its inhibitor rapamycin. This last result confirms the senolytic property of the mTOR inhibitors [32]. At the state of the art, rapamycin is the only drug approved for the treatment of LAM that has a cytostatic and not cytotoxic action, and discontinuation of the treatment leads to tumour regrowth and decline in pulmonary function [33,34,35]. Considering a possible senescent behaviour of LAM cells, the relapse after rapamycin treatment suspension might be due to the reactivation of the so-called geroconversion, an irreversible state of senescence characterized by cellular growth sustained by mTOR. Indeed, we observed that the inhibition of mTOR by rapamycin reduces the phosphorylation of ZRF1, which drives the mTOR-dependent senescent programme [23], without changes in p21^WAF1/CIP1^ expression, indicating that LAM/TSC cells are senescent in spite of a state of cell cycle arrest. As further confirmation of the role of mTOR hyperactivation in sustaining the senescent phenotype of LAM/TSC cells, we obtained similar results on phospho-ZRF1 and p21^WAF1/CIP1^ expression when LAM/TSC cells were treated with 5-azacytidine, which causes tuberin expression. Since 5-azacytidine is a methyltransferase inhibitor, this drug is expected to induce senescence in a p53 dependent manner as reported for oral squamous cell carcinoma, hepatocellular carcinoma cells, and other solid tumour cell lines [36]. Conversely, our results indicate that the treatment with 5-azacytidine, inducing tuberin expression, reduced the senescent features of LAM/TSC cells, which decrease the positivity to SA-β galactosidase. These data are consistent with the recent study in which 24 h of incubation with 5-azacytidine inhibited senescence in insulinoma cell lines established from transgenic mice [37]. In this case, the drug treatment caused cytotoxicity, which was not observed in our experimental conditions. We previously demonstrated that 5-azacytidine treatment also reduced migratory capability and MMP2 and MMP7 secretion of LAM/TSC cells, likely because of the induction of tuberin expression [11]. Given the broad inhibitory action of 5-azacytidine on hypermethylation that leads to chromatin remodelling, as often occurs in several types of cancers, we have to consider that this drug might cause other demethylating effects besides the induction of tuberin expression, making this point a limitation of our study that could be overcome by the retroviral transduction of *TSC2*. However, all the effects of 5-azacytidine that we know to occur in our model are not supposed to influence what we observed as reduction of senescence features and, considering together the rapamycin and 5-azacytidine action, we may reasonably affirm that LAM/TSC cell senescent features are caused by mTOR hyperactivation due to the absence of tuberin. In the controversial discussion of the role of senescence in cancer, considering the neoplastic characteristics of LAM cells, these data indicate that cancer cells are in fact pro-senescent and capable to promote senescence, as shown in cells with loss of PTEN [38].

LAM/TSC cells survive in an anchorage-independent manner, degrade the extracellular matrix through the secretion of matrix metalloproteinases, suggesting a migratory ability, and secrete high levels of IL-6 and IL-8 [16]. All this evidence also demonstrated the capability of these cells to create a supporting microenvironment for their survival in the lung. To model the possible communication between LAM/TSC cells and PLFs (not by means of a direct contact), PLFs were grown in the CM of LAM/TSC cells. PLFs proliferation was significantly reduced probably because of the activation of p21^WAF1/CIP1^ protein that induces cell cycle arrest without increasing cellular apoptosis, as shown by the annexin V results and expression of Caspase 3 and PARP. In this condition, also a significantly higher activation of SA-β-galactosidase and higher phosphorylation of histone H2A.X were observed, confirming a senescent behaviour of PLFs induced by LAM/TSC cell CM but not by their own CM. The LAM/TSC cell CM also caused senescent features in IMR90 cell line sustaining the capability of LAM/TSC cells to induce senescence in fibroblasts. Very recently, the capability of LAM cells to modulate the LAM-Associated fibroblasts (LAFs) phenotype was demonstrated in an in vitro model of LAM microtissues, where non-expressing-tuberin LAM cells cause the increased expression of the Fibroblasts Growth Factor 7 protein, which acts as a chemotactic and mitogen factor for epithelial cells [18]. This evidence sustains the role of the LAM microenvironment in controlling LAM development.

IL-8 is a potent neutrophil chemotactic factor that plays important roles under several pathological and physiological conditions including senescence, which mediates the spreading of the SASP response in the microenvironment [6]. We demonstrated that LAM/TSC cells not only express and secrete more IL-8 than PLFs, but they are also able to cause a significant increase of IL-8 mRNA levels and IL-8 secretion in PLFs grown in LAM/TSC cell CM, while PLF CM does not cause any alteration in IL-8 expression and secretion. This evidence suggests that, in LAM, senescent LAM cells might reduce the proliferation of lung resident cells and, at the same time, might promote a pro-inflammatory microenvironment that ultimately has detrimental effects on the lung tissue. Indeed, even if IL-8 has a clear role as a SASP molecule, its secretion by fibroblasts, alveolar macrophages, and alveolar cells and its involvement in neutrophil migration and infiltration in lungs suggest a more complex physiological implication, making the deregulation of IL-8 secretion responsible for pathological processes, i.e., in IPF [25], acute respiratory distress syndrome [26], and chronic obstructive pulmonary disease [39]. Besides the crucial role of IL-8 in lung inflammation, little is known concerning its involvement in LAM or TSC. We observed that high IL-8 secretion occurs in LAM/TSC cells, and it is decreased after the treatment with 5-azacytidine indicating that its production in these cells might be dependent on tuberin-sensitive pathways [16]. However, the fact that IL-8 secretion is increased in PLFs when grown in LAM/TSC cell CM and the evidence that in this condition PLFs become senescent suggest that IL-8 secretion in PLFs occurs in a senescence-dependent manner. In LAM or TSC models, the SASP cytokine IL-6 has been better studied than IL-8, demonstrating an increase in TSC2-deficient cells in an mTORC1-dependent manner and an up-regulation in plasma of LAM patients compared to healthy subjects [28]. In the in vitro model, IL-6 is involved in migration and proliferation as they are inhibited by the treatment with anti-IL-8 antibody. Similar to IL-6, the IL-8 high secretion of LAM/TSC cells might be responsible of other processes than senescence in LAM microenvironment.

Concluding, the demonstration that LAM cells are senescent and have the capability to induce senescence in neighbouring cells introduces an interesting approach to investigate novel targets to control LAM progression. Indeed, to understand the involvement of mTOR and of its specific age-related targets in LAM senescent cells might help to better evaluate the effect of rapamycin in treating LAM. Finally, interfering with IL-8 expression and secretion may help to conceive a novel approach to delay the progression of the lung parenchyma disruption.

## 4. Materials and Methods

### 4.1. Cell Culture and Treatments

LAM/TSC cells were previously obtained from the chylous thorax of an LAM/TSC2 patient who had given written informed consent according to the Declaration of Helsinki [16]. The study was approved by the Institutional Review Board of Milan’s San Paolo Hospital. As previously reported, LAM/TSC cells can be grown as a stabilized cell line in Type II medium, composed by a 1:1 mixture of stable Glutamine-containing DMEM/Ham’s F12 (Aurogene s.r.l, Rome, Italy) supplemented with 200 nmol/L hydrocortisone (Sigma-Aldrich, St. Louis, MO, USA), 10 ng/mL epidermal growth factor (EGF; Sigma-Aldrich), 100 µg/mL penicillin/streptomycin (Sigma-Aldrich), and 15% foetal bovine serum (Sigma-Aldrich). LAM/TSC cells were routinely checked for morphological, biochemical, and genetic features.

Human Primary Lung Fibroblasts (PLFs) were obtained from a lung biopsy. The tissue was washed in PBS, cut into small pieces, and placed in cell culture dishes with complete Type II medium for 3–5 days. When the fibroblasts reached the confluency, they were sub-cultured as a stabilized cell line.

In order to avoid non-specific effects due to aging cells that could be confused with senescence, all the experiments with the primary cells were performed within 20 days from cell thawing, meaning between passage 3 to 5.

IMR90 cells, a human foetal lung fibroblast cell line (ATCC, Rockville, MD, USA), were maintained in stable Glutamine-containing DMEM supplemented with 10% FBS and 100 µg/mL penicillin/streptomycin. To prevent the onset of a senescent phenotype due to cell aging, IMR90 cells were used within a population doubling level (PDL) 20 to 30. In this PDL range, IMR90 cells duplicate at a constant rate, indicating that senescence process is not still begun, while at PDL levels over 40, a growing delay starts (data not shown). PDL was calculated with the following formula: PDL = [PDL_0_ + 3.22 × (Log_10_ CN_Y_ − Log_10_ CN_P_)] where PDL_0_ is the initial PDL, CN_Y_ is the number of cells yielded, and CN_P_ is the number of cells plated.

The CM was obtained from LAM/TSC cells and PLFs grown in standard condition for 72 h and pre-cleared from suspended cells and debris by centrifugation at 400× *g* for 5 min at room temperature. PLFs or IMR90 cells were grown for 48 h in LAM/TSC cell CM or in their own CM as control.

LAM/TSC cells were treated with rapamycin (0.5 ng/mL and 1 ng/mL; Rapamune-Sirolimus, Wyeth Europa, Sandwich, Kent, UK) for 48 h or with 5-azacytidine (1 µM, Sigma-Aldrich) for 96 h.

Proliferation assay was performed by plating 5 × 10^4^ PLFs in CM of LAM/TSC cells or in their own CM, as control. The cells were harvested and counted after 24, 48, 72, and 96 h using a Neubauer chamber. Cells viability was evaluated with trypan blue exclusion assay as previously described [16].

### 4.2. Western Blot Analysis

Western Blot (WB) analysis was performed as previously described [11]. Briefly, cells were lysed in lysis buffer (5 mM EDTA, 100 mM deoxycholic acid, 3% sodium dodecyl sulphate supplemented with protease inhibitors: Benzamidine 1 mM, PMSF 400 µM, Leupeptine 1 µg/mL, Aprotinin 10 µg/mL, Sigma-Aldrich), boiled for 5 min, and 30 µg of proteins for each sample were analysed by SDS-PAGE followed by the transfer to nitrocellulose membranes (Amersham, Arlington Height, IL, USA). After blocking at room temperature for 1 h with 5% dry milk (Merck, Darmstadt, Germany), membranes were incubated overnight at 4 °C with antibodies against Tuberin (1:500; Cell Signalling Technologies, Danvers, MA, USA, Cat. No. 3635), p16^INK4A^(D7C1M) (1:1000; Cell Signalling Technologies, Cat. No. 80772), phospho-S6 (Ser235/236) (1:1000; Cell Signalling Technologies, Cat. No. 2211), p21^WAF1/CIP1^(12D1) (1:1000; Cell Signalling Technologies, Cat. No. 2947) phospho-ZRF1 (DNAJC2/MPP11) (Ser47) (1:1000; Cell Signalling Technologies, Cat. No. 12397), Caspase 3 (8G10) (1:500; Cell Signalling Technologies, Cat. No. 9665), PARP-1 (H250) (1:100; Santa Cruz Biotechnology, Dallas, TX, USA, Cat. No. 7150), or β-actin (1:1000; Sigma-Aldrich, Cat. No. A5441). The appropriate secondary horseradish peroxidase conjugate antibodies (1:10,000; Thermo Scientific, Rockford, IL, USA, Cat. No. 31430/31460) were incubated for 1 h at room temperature, and the reaction was revealed by using the ECLT Prime Western Blotting System (Amersham, UK) or WesternBright Sirius HRP Substrate (Advansta, San Jose, CA, USA). Images were acquired on a Kodak image station 1550 GL. Densitometric analysis of each band relatively to β-actin levels was performed in at least three independent experiments by using Kodak 1D 3.6 Software Image (Kodak, Milano, Italy).

### 4.3. SA-βGalactosidase Assay

In total, 7 × 10^4^ LAM/TSC cells, PLFs or IMR90 cells were seeded, grown as indicated for each experiment, and stained using the Senescence β-Galactosidase Staining Kit (Cell Signalling Technology), according to manufacturer’s instruction. Images were acquired with a bright-field microscope, and senescent cells were evaluated as the percentage of the stained blue senescent cells over the total number of cells by counting at least 10 fields for each sample in three independent experiments.

### 4.4. Flow Cytometric Analysis

The histone H2A.X Phosphorylation assay Kit (Sigma Aldrich, Cat. No. 17-344) was used according to manufacturer’s instruction. Briefly, LAM/TSC cells, PLFs, and IMR90 cells, grown as reported for each experiment, were collected, washed in PBS, and fixed in 2% paraformaldehyde for 20 min on ice. Direct labelling was performed by incubating the cells with the antibody for histone H2A.X FITC-conjugated or normal mouse IgG isotype control FITC-conjugated (0.3 µg for each sample) in an appropriate volume of Permeabilization Solution for 20 min on ice. Finally, samples were washed with Wash Buffer.

For apoptosis evaluation, PLFs grown in LAM/TSC cell CM, in their own CM, or treated with 500 µM H_2_O_2_ for 2 h were detached with Accutase (Sigma-Aldrich), washed in PBS, and incubated for 15 min at room temperature with APC Annexin V (BD Biosciences, Franklin Lakes, NJ, USA, Cat. No. 550475) in an appropriate volume of Annexin V Binding Buffer (BD) and washed in PBS.

Samples were analysed by Cytomics FC500 (Beckman Coulter, Brea, CA, USA). Data acquisition and analysis were done using CXP 2.2 software (Beckman Coulter).

### 4.5. ELISA

The total amount of IL-8 in cell culture media was measured by Enzyme-Linked Immunosorbent Assay (ELISA) kit (Cat. No. IL-8: SEA080Hu-96, Cloud-Clone Corp, Katy, TX, USA) according to the manufacturer’s instruction.

### 4.6. qPCR

Total RNA was extracted from LAM/TSC cells or PLFs, grown in standard condition or in the CM, using NucleoZOL (Macherey-Nagel, Düren, Germany) and phenol-clorophorm method. cDNA was synthesized from 1 μg of total RNA using the High-Capacity cDNA Reverse Transcription Kit (Applied Biosystems, Thermo Fisher Scientific) and random primers. qPCR was performed on a Roche LightCycler 480 qPCR machine using precision TB Green Premix Ex TaqII (Tli RNaseH Plus, Takara Bio, Saint-Germain-en-Laye, France) qPCR Master Mix with SYBR Green (Takara Bio, Saint-Germain-en-Laye, France). First-strand DNA synthesis reactions without reverse transcriptase were used as controls. Triplicate samples are run in all reactions. The quantification cycle (ΔCq) value and the ΔΔCq were calculated relative to control samples using Cq threshold values that are normalized to the housekeeping gene *GAPDH*, *RPLP0*, *RPL13A*.

The primers used were as follows:

h_*GAPDH*, forward (fw), 5′-agccacatcgctcagacac-3′;

reverse (rv), 5′-gcccaatacgaccaaatcc-3′;

h_*RPLP0*, fw, 5′-tctacaaccctgaagtgcttgat-3′;

rv, 5′-caatctgcagacagacactgg-3′;

h_*RPL13A*, fw, 5′-CCTGGAGGAGAAGAGGAAAGAGA-3′;

rv, 5′-TTGAGGACCTCTGTGTATTTGTCAA-3′;

h_*IL8*, fw, 5′-TTTCCACCCCAAATTTATCAAAG-3′;

rv, 5′-CAGACAGAGCTCTCTTCCATCAGA-3′

### 4.7. Statistical Analysis

Statistical analysis was performed with the GraphPad Prism 7.02 software. (GraphPad Software, San Diego, CA, USA). Differences between the mean measurement of each sample were analysed using Student’s *t*-test for single comparison or one-way Analysis of Variance (ANOVA) followed by the Tukey’s test for multiple comparisons, when appropriate. Results are presented as mean ± standard error of the mean (SEM). *p* values were considered significant as reported in each figure.

## Figures and Tables

**Figure 1 ijms-23-07040-f001:**
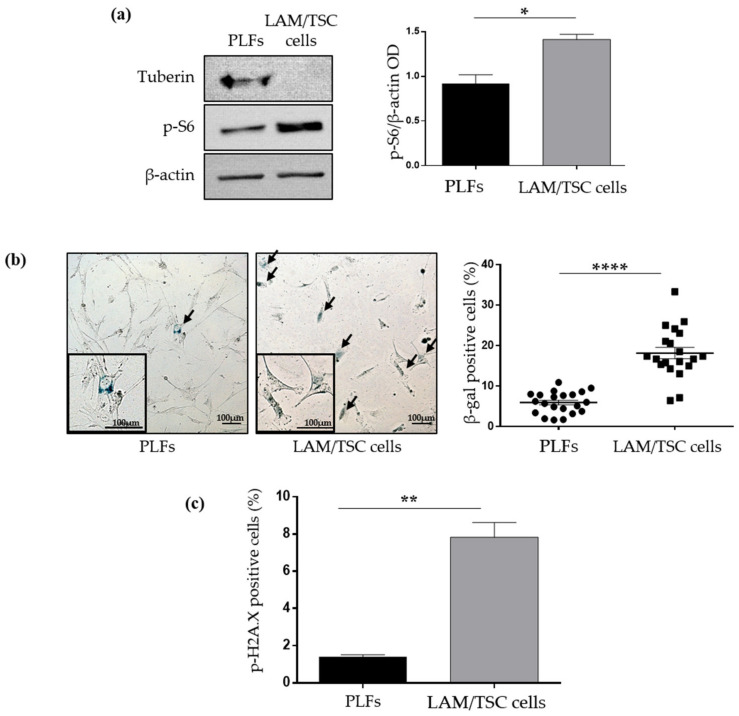
Analysis of senescent features in PLFs and LAM/TSC cells. (**a**) Left panel: expression of tuberin and levels of phosphor-S6, mTOR substrate, were analysed by western blot in PLFs and LAM/TSC cells. Β-actin was used as loading control. Representative images are shown. Right panel: densitometric analysis of p-S6/β-actin. Results are shown as mean ± SEM (*n* = 4). Student’s *t*-Test, * *p* < 0.05. (**b**) Representative images of SA-β-Galactosidase staining of PLFs (**left**) and LAM/TSC cells (**right**) are shown. Senescent cells (blue cells) are highlighted by arrows. Higher magnifications of senescent cells are shown in black squares. Scale bar: 100 µm. The dot plot represents the percentage of SA β-galactosidase positive cells obtained by counting the positive cells on the total number of cells in 7 images of 3 independent experiments. Results are shown as mean ± SEM (*n* = 21). Student’s *t*-Test, **** *p* < 0.0001. (**c**) Histograms display the percentage of PLFs and LAM/TSC cells positive to phospho-histone H2A.X (p-H2A.X) evaluated by flow cytometry analysis. Results are shown as mean ± SEM, *n* = 3 experiments, Student’s *t*-Test, ** *p* < 0.01.

**Figure 2 ijms-23-07040-f002:**
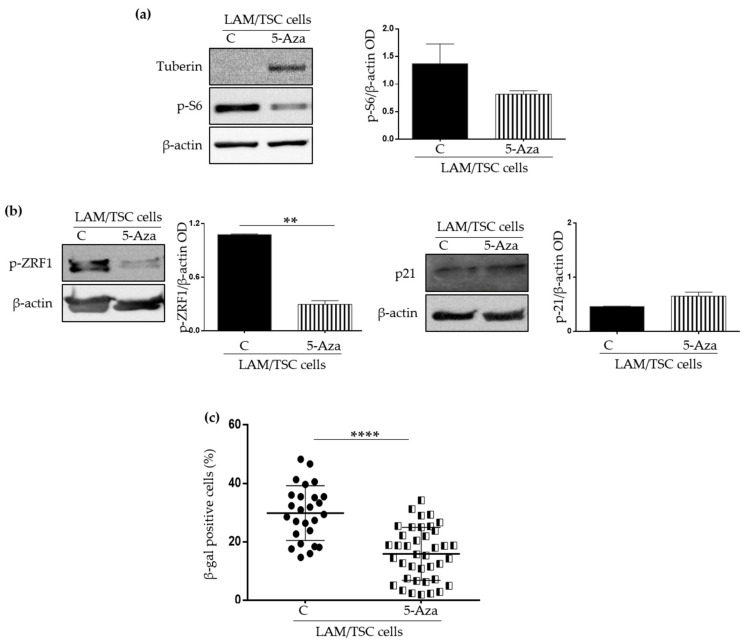
Analysis of the senescent characteristics of LAM/TSC cells following the induction of tuberin expression by 5-Azacytidine. (**a**) Tuberin and phospho-S6 (p-S6) expression were analysed by Western Blot analysis in LAM/TSC cells (C) and in LAM/TSC cells treated with 1 µM 5-Azacytidine (5-Aza) for 96 h. Representative images of 3 independent experiments are shown. β-actin was used as loading control. Right panel: densitometric analysis of p-S6/β-actin. Results are shown as mean ± SEM (*n* = 3). Student’s *t*-Test. (**b**) Senescent markers p21^WAF1/CIP1^ (p21) and phospho-ZRF1 (p-ZFR1) were analysed by Western Blot in LAM/TSC cells (C) and in LAM/TSC cells treated with 5-Azacytidine for 96 h (5-Aza). Representative images of 3 independent experiments are shown. β-actin was used as loading control. Densitometric analysis of p-ZRF1/β-actin and p21/β-actin are presented. Results are shown as mean ± SEM (*n* = 3). Student’s *t*-Test, ** *p* < 0.01 (**c**) Quantification of β galactosidase-positive LAM/TSC cells, considered as the percentage of stained cells on the total number of cells in each field, was performed on 3 independent experiments in standard condition (C) or after the treatment with 5-Azacytidine for 96 h (5-Aza). 8 images for each experiment were evaluated. Results are shown as mean ± SEM (*n* = 24). Student’s *t*-Test, **** *p* < 0.0001.

**Figure 3 ijms-23-07040-f003:**
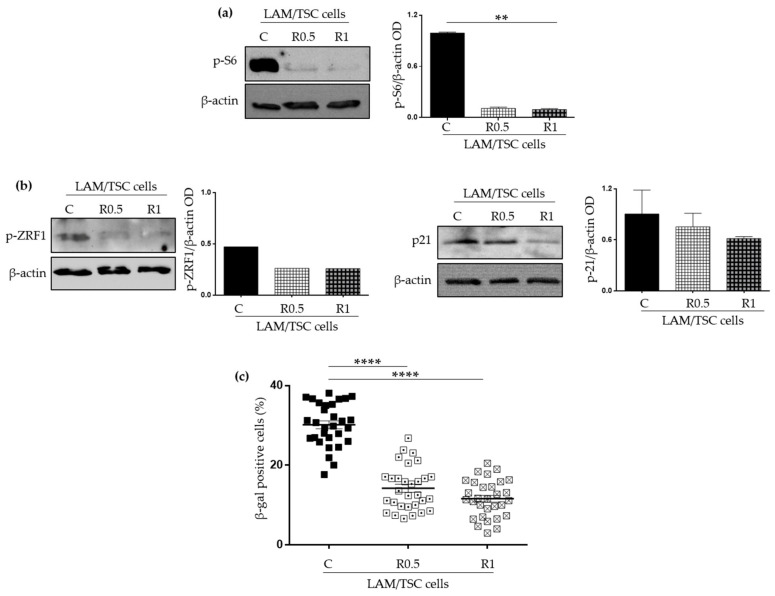
Analysis of the senescent characteristics of LAM/TSC cells following mTOR inhibition by rapamycin. (**a**) Phospho-S6 (p-S6) levels were evaluated in LAM/TSC cells (C) and LAM/TSC cells treated for 48 h with 0.5 ng/mL (R0.5) or 1 ng/mL (R1) of rapamycin. β-actin was used as loading control. Densitometric analysis of p-S6/β-actin is shown. Results are shown as mean ± SEM (*n* = 3). ANOVA with Tukey Test, ** *p* < 0.01 (**b**) Senescent markers p21^WAF1/CIP1^ (p21) and phospho-ZRF1 (p-ZRF1) were analysed by Western Blot analysis in LAM/TSC cells (C) and in LAM/TSC cells treated for 48 h with 0.5 ng/mL (R0.5) or 1 ng/mL (R1) of rapamycin. β-actin was used as loading control. Densitometric analysis of p-ZRF1/β-actin and p21/β-actin are shown. Results are shown as mean ± SEM (*n* = 3). ANOVA with Tukey, *ns* (**c**) Quantification of SA-β galactosidase positive LAM/TSC cells was performed in standard condition (C) or after the treatment with 0.5 ng/mL (R0.5) or 1 ng/mL (R1) of rapamycin for 48 h on 3 independent experiments evaluating 10 images each experiment. Results are shown as mean ± SEM (*n* = 30). ANOVA with Tukey Test, **** *p* < 0.0001.

**Figure 4 ijms-23-07040-f004:**
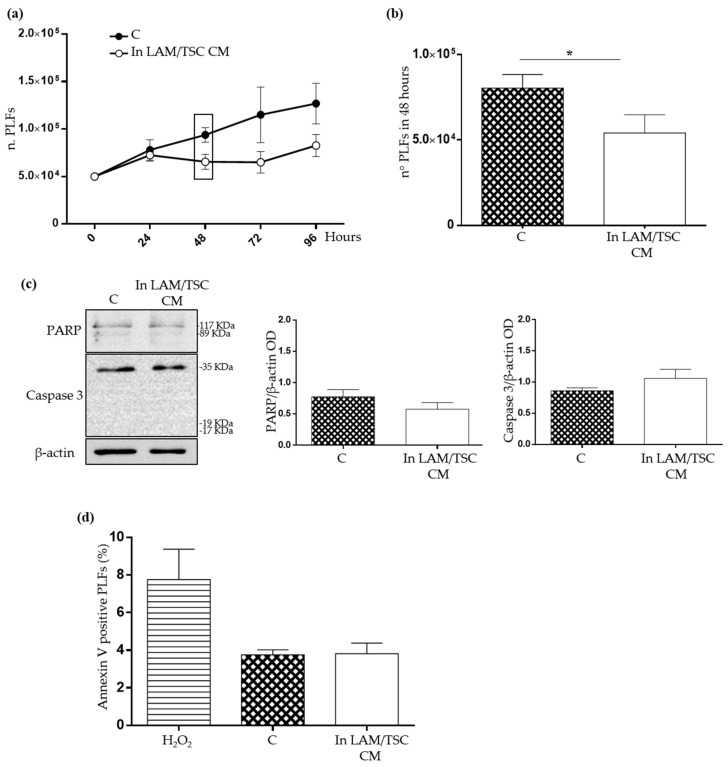
Proliferation of PLFs in LAM/TSC cell CM (**a**) Growth of PLFs cells was evaluated at 24, 48, 72, and 96 h in their own CM (C) or in LAM/TSC cell CM (LAM/TSC CM). The rectangle indicates the 48 h-time point. Each data point shows the mean ± SEM (*n* = 4). (**b**) The histogram highlights the number of PLFs (C) and PLFs in LAM/TSC cell CM after 48 h of growth. Results are shown as mean ± SEM (*n* = 4). Student’s *t*-Test, * *p* < 0.05 (**c**) PARP and Caspase 3 expression was analysed in PLFs cells grown for 48 h in their own CM (C) and in LAM/TSC cell CM. β-actin was used as loading control. Anti-Caspase 3 antibody and anti-PARP antibody recognize both the full length and the cleaved form of the protein (the molecular weight of the full-length protein and of its expected cleaved fragments are indicated on the right of the image). Representative images of 3 independent experiments are shown. The histograms on the right represent the densitometric analysis of the relative expression of PARP and Caspase 3 normalized on β-actin. Results are shown as mean ± SEM (*n* = 3). Student’s *t*-Test, *ns* (**d**) Flow cytometry analysis of annexin positive PLFs cells was performed in cells grown for 48 h in their own CM (C) or in LAM/TSC cell CM. The induction of apoptosis by treating PLFs with 500 µM Hydrogen Peroxide (H_2_O_2_) for 2 h was used as positive control. Results are shown as mean ± SEM (*n* = 3).

**Figure 5 ijms-23-07040-f005:**
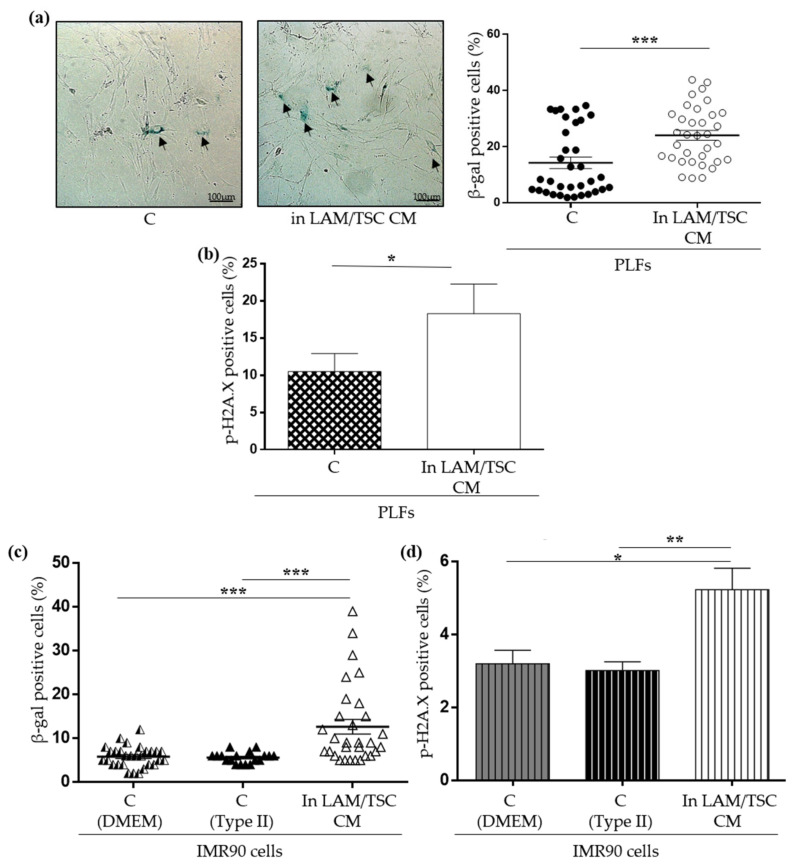
Senescence effect of LAM/TSC cell CM on PLFs and IMR90 cells. (**a**) Representative images of SA-β-Galactosidase staining of PLFs grown in their own CM (C, **left**) and in LAM/TSC cell CM (**right**) are shown. The arrows indicate the senescent cells (blue cells). Scale bar: 100 µm. The dot plot represents the % of β galactosidase positive cells evaluated in 10 images of 3 independent experiments. Results are shown as mean ± SEM (*n* = 3). Student’s *t*-Test, *** *p* < 0.001. (**b**) Flow cytometry analysis of phospho-histone H2A.X positive PLFs cells grown in their own CM (C) and in LAM/TSC cell CM are expressed as percentage of the total number of cells. Results are shown as mean ± SEM (*n* = 3). Student’s *t*-Test, * *p* < 0.05. (**c**) The dot plot shows the percentage of SA β-galactosidase positive IMR90 cells grown in standard condition (C-DMEM), in Type II medium (C-Type II) or in LAM/TSC cell CM for 48 h. 10 images of 3 independent experiments were evaluated. Results are shown as mean ± SEM (*n* = 30). ANOVA with Tukey Test, *** *p* < 0.001. (**d**) Flow cytometry analysis of phospho-histone H2A.X positive IMR90 cells was performed in cells grown in standard condition (C-DMEM), in Type II medium (C-Type II) and in LAM/TSC cell CM for 48 h. Results are expressed as percentage and shown as mean ± SEM (*n* = 3). ANOVA with Tukey Test, * *p* < 0.05, ** *p* < 0.01.

**Figure 6 ijms-23-07040-f006:**
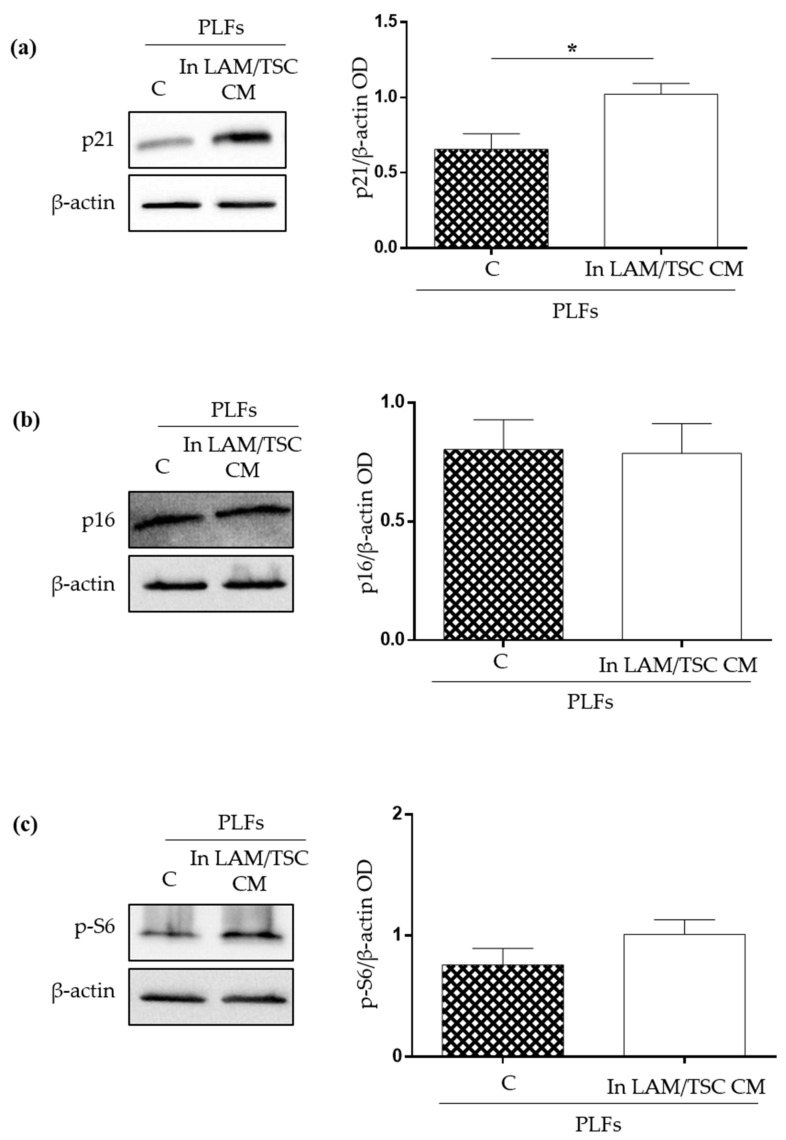
Evaluation of the effects of LAM/TSC cell CM on p21^WAF1/CIP1^ and p16^INK4A^ expression and on S6 phosphorylation in PLFs. (**a**) p21^WAF1/CIP1^ (p21) expression was analysed in PLFs grown for 48 h in their own CM (C) or in LAM/TSC cell CM. β-actin was used as loading control. Right panel: densitometric analysis of p21^WAF1/CIP1^/β-actin. Results are shown as mean ± SEM (*n* = 3). Student’s *t*-Test, * *p* < 0.05. (**b**) p16^INK4A^ (p16) expression was studied in PLFs grown for 48 h in their own CM (C) or in LAM/TSC cell CM. β-actin was used as loading control. Right panel: densitometric analysis of p16^INK4A^/β-actin. Results are shown as mean ± SEM (*n* = 3). Student’s *t*-Test, *ns*. (**c**) The phosphorylation of S6 (p-S6) was analysed in PLFs grown for 48 h in their own CM (C) or in LAM/TSC cell CM. β-actin was used as loading control. Right panel: densitometric analysis of p-S6/β-actin. Results are shown as mean ± SEM (*n* = 3). Student’s *t*-Test, *ns*.

**Figure 7 ijms-23-07040-f007:**
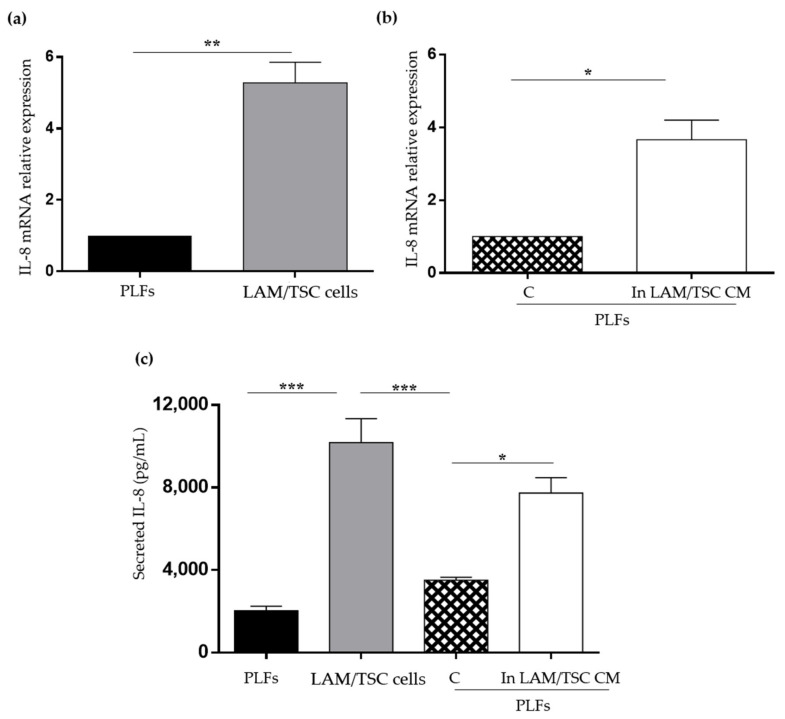
Analysis of IL-8 mRNA expression and secretion in PLFs and LAM/TSC cells, and in PLFs in CM. (**a**) mRNA relative expression of IL-8 was evaluated in PLFs and LAM/TSC cells. Results are shown as mean ± SEM (*n* = 3). Student’s *t*-Test, ** *p* < 0.01. (**b**) mRNA relative expression of IL-8 was analysed in PLFs grown in their own CM (C) and in LAM/TSC cell CM. Results are shown as mean ± SEM (*n* = 3), Student’s *t*-Test, * *p* < 0.05. (**c**) 48 h-IL-8 secretion was analysed by ELISA in PLF and LAM/TSC cell supernatants (columns 1 and 2), and in the supernatant of PLFs grown in their own CM (C) and in LAM/TSC cell CM (columns 3 and 4). The IL-8 levels in columns 3 and 4 are obtained by the subtraction of the IL-8 measured in the supernatant of PLFs (column 1) and in LAM/TSC cells (column 2) to the IL-8 measured in the PLFs grown in their own CM or in LAM/TSC cell CM, respectively. A total of 150,000 PLFs or LAM/TSC cells were plated for each sample. Results are shown as mean ± SEM, *n* = 3, ANOVA with Tuckey Test, * *p* < 0.05, *** *p* < 0.001.

## Data Availability

Not applicable.

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
