# Peer review of "LAM Cells as Potential Drivers of Senescence in Lymphangioleiomyomatosis Microenvironment"

_ijms, 2022, doi:10.3390/ijms23137040_

Round 1
Reviewer 1 Report
Bernardelli and colleagues examine whether senescence is a driving factor in the LAM phenotype. They demonstrate that LAM cells have senescent features, including SA-β galactosidase activity, p21 levels, phosphorylated histone H2A.X, and high levels of IL-8 expression and secretion. The utilized 5-azacytidine to trigger TSC2 expression and LAM cell condition media to treat human lung fibroblasts. They observed that TSC2 expression could reverse several senescent features, and LAM conditioned media could enhance these features in the fibroblasts. Overall, this is an important topic and novel manuscript. Here are the areas that need addressing, in no particular order:
1. I would recommend rewriting the abstract, as a reader I did not know the methodology approach and primary outcomes of the study while reading the abstract.
2. While IL8 secretion is a known SASP, can higher levels of IL8 occur independently of senescence in this model?
3. Was the conditioned media the same composition as the controlled media for normal fibroblast culture or IMR90 cells? In the methods, it appears that you use DMEM/F12 for LAM cells and DMEM for fibroblasts. Are the supplements the same? Several components of the media alone might influence senescence, such as glucose, insulin levels, serum source, or concentrations (e.g., Maeda et al., Plos One, 2015, https://doi.org/10.1371/journal.pone.0123169 or Matsui-Hirai et al., J Pharmacol Exp Ther. 2011, https://doi.org/10.1124/jpet.110.177584, or Duggal, et al., J Cell Physiol. 2011, doi: 10.1002/jcp.22637)
4. Please include cell passage numbers in the methodology as aging cells are more prone to becoming senescent
5. Throughout the manuscript, please quantify Western data by densitometry
6. Please include either viability or cytotoxicity data following treatment with 5-azacytidine and rapamycin
7. It would be a cleaner experiment to reintroduce TSC2 expression using vectors rather than using 5-azacytidine or rapamycin. What other none specific effects would be expected using this compound? Please include this in a limitation portion of the discussion
8. Large portions of the discussion read as an introduction, e.g., lines 277-303. The opening paragraph should outline the main findings from this study and discuss the relevance to the field
Author Response
- I would recommend rewriting the abstract, as a reader I did not know the methodology approach and primary outcomes of the study while reading the abstract.
We thank the Reviewer for the recommendation to make the abstract more suitable. We rewrote it, pointing out the methodology approach and the primary outcomes. We hope that now the abstract results clearer for the readers.
- While IL8 secretion is a known SASP, can higher levels of IL8 occur independently of senescence in this model?
IL-8 has several roles mainly related to inflammation, besides being a SASP factor. To our knowledge, in LAM or TSC models, there are no reported evidence of the effects of high IL-8 levels or of the induction of high IL-8 secretion. However, only as comparison there are data regarding IL-6, a SASP cytokine, that is demonstrated to be important for proliferation and migration of tuberin-null cells (reference 28). Moreover, IL-6 is upregulated in plasma of LAM patients. We can suppose that similarly also IL-8 is involved in LAM/TSC cell processes besides senescence. In discussion this point has been introduced (line 421-433)
- Was the conditioned media the same composition as the controlled media for normal fibroblast culture or IMR90 cells? In the methods, it appears that you use DMEM/F12 for LAM cells and DMEM for fibroblasts. Are the supplements the same? Several components of the media alone might influence senescence, such as glucose, insulin levels, serum source, or concentrations (e.g., Maeda et al., Plos One, 2015, https://doi.org/10.1371/journal.pone.0123169 or Matsui-Hirai et al., J Pharmacol Exp Ther. 2011, https://doi.org/10.1124/jpet.110.177584, or Duggal, et al., J Cell Physiol. 2011, doi: 10.1002/jcp.22637)
We thank the Reviewer for the observation that highlights an important point. In standard condition LAM/TSC cells grow in DMEM/Ham’s F12 (1:1) with 15% FBS (1.1; Type II medium) while IMR90 cells in DMEM with 10% FBS. In the first submitted version, the LAM/TSC cell CM added to IMR90 cells was the Type II medium to better compare the CM effect obtained in PLFs. Considering the crucial role of the media components for the modulation of senescence (as demonstrated also in the articles cited by the Reviewer, reference 24 has been added), we previously performed experiments on IMR90 cells comparing the effect of DMEM-IMR90 cell CM and Type II-IMR90 cell CM, without obtaining any significant differences in the senescent hallmarks. The fact that Type II medium is DMEM/Ham’s F12 in a dilution 1:1 and CM is an exhausted medium likely reduces the differences of the medium components crucial for the induction of senescence in our experimental conditions. It will be interesting to better investigate some of the key media components, such as glucose and various FBS concentrations, to understand their role in inducing senescence in LAM/TSC cells. Giving the lack of significant differences using DMEM and Type II medium in IMR90 cells, in the first submission, we wrongly underestimated the importance of showing the results obtained in studying senescence. Therefore, following the reviewer’s request, these data are now reported in Figure 5c and d (line 247-260).
- Please include cell passage numbers in the methodology as aging cells are more prone to becoming senescent
The cell passage numbers for IMR90 cells and LAM/TSC cells have been included. We used IMR90 cells between 20 to 30 passage numbers and LAM/TSC cells between 3 to 5 (line 457-458 and 462-467).
- Throughout the manuscript, please quantify Western data by densitometry
The western blot data has been quantified by densitometry and the results have been added to Figure 1a, Figure 2a and b, Figure 3a and b, and Figure 4c.
- Please include either viability or cytotoxicity data following treatment with 5-azacytidine and rapamycin
The study of viability and cytotoxicity after 5-azacytidine and rapamycin treatment in LAM/TSC cells have been performed and added in the Supplemental Figure 1 and 2 (line 155-160 and 185-192). Cell viability was evaluated with trypan blue exclusion assay showing no significant difference between control cells and the groups treated with 0.5 ng/mL and 1ng/mL rapamycin for 48 hours, or 1 µM 5-azacytidine for 96 hours confirming our previous data (reference 16). The drugs did not cause apoptosis as shown by the absence of fragments of caspase 3 and PARP analysed by western blot and by any difference for Annexin V positivity between control and treated groups, which also demonstrates no significant presence of necrotic cells in our experimental conditions.
- It would be a cleaner experiment to reintroduce TSC2 expression using vectors rather than using 5-azacytidine or rapamycin. What other none specific effects would be expected using this compound? Please include this in a limitation portion of the discussion
We thank the reviewer for the request that aims to give a comment on an important issue missing in first submitted discussion. It has been demonstrated that high concentrations of 5-azacytidine have cytotoxic effects and lower doses have demethylating activity. As discussed in the revised version of the manuscript and in point 3, the used concentration does not cause significant cytotoxic effects. However, the demethylating activity can have multiple targets besides the epigenetic modification in TSC2. This is a limitation for our model that could be only overcome by the retroviral transduction of the TSC2 gene even if the induction of tuberin expression by the chromatin remodelling agent is considered specific for this epigenetic modification. We introduced this concept in the discussion (line 363-384, references 36 and 37 have been added).
- Large portions of the discussion read as an introduction, e.g., lines 277-303. The opening paragraph should outline the main findings from this study and discuss the relevance to the field
The initial portion of the discussion has been revised, the part similar to the introduction has been removed and as requested, an opening paragraph has been included. We hope that in this version the discussion is now acceptable.

Reviewer 2 Report
The manuscript by Bernardelli et al. describes effect of Lymphangioleiomyomatosis (LAM) senescence on surrounding cells via senescence-associated secretory phenotype (SASP). The authors found high levels of senescence in LAM cells. 5-Aza and rapamycin treatments decreased levels of p-S6 and percentages of beta-gal positive cells. The conditioned medium (CM)of LAM inhibited cell growth in pulmonary lung fibroblasts (PLF). In addition to this, the LAM CM induced senescence phenotype in IMR90 cells. IL-8 mRNA and protein levels significantly increased in PLFs after the LAM CM treatment. The manuscript showed an interesting function of LAM CM, which induced senescence phenotype in PLF. However, there are several minor concerns.
1. In abstract, the authors mentioned “in which mTOR is not hyperactivated”. Have the authors examined if LAM CM induced mTOR hyperactivation in PLFs? If so, it would be better to show the data.
2. In materials and methods, the authors mentioned that ANOVA followed by the Tukey’s test was used for multiple comparisons. However, in the analyses, Student’s t-test was used for multiple comparison (at least shown in the figure legends). For example, Figure 3C and Figure 7C.
3. In Figure 7C, LAM CM contained IL-8, so how did the authors measure IL-8 levels in PLFs with LAM CM? it might be simply measuring IL-8 in LAM CM, not secreted from PLFs.
Author Response
- In abstract, the authors mentioned “in which mTOR is not hyperactivated”. Have the authors examined if LAM CM induced mTOR hyperactivation in PLFs? If so, it would be better to show the data.
The mTOR activation detected as phosphorylation of the substrate S6 was previously analysed following the treatment with LAM/TSC cell CM and now added to the Figure 6 (Figure 6c) (line 282-283) following the suggestion of the Reviewer. We did not wrongly include this data in the first version. The LAM/TSC cell CM very slightly increased P-S6. Indeed, the quantification by densitometry of the western blot data resulted not statistically significant.
- In materials and methods, the authors mentioned that ANOVA followed by the Tukey’s test was used for multiple comparisons. However, in the analyses, Student’s t-test was used for multiple comparison (at least shown in the figure legends). For example, Figure 3C and Figure 7C.
We truly apologize for the mistake in writing the legends of Figure 3C and 7C. As correctly observed by the reviewer, in these Figures the statistical analysis was performed with ANOVA followed by the Tukey’s test for multiple comparison and NOT with Student’s t-test (that was used for single comparison). The legends 3C and 7C have been corrected.
- In Figure 7C, LAM CM contained IL-8, so how did the authors measure IL-8 levels in PLFs with LAM CM? it might be simply measuring IL-8 in LAM CM, not secreted from PLFs.
We thank the Reviewer for the observation since we probably did not clearly explain how the IL-8 levels have been measured and analyzed. The IL-8 secretion was evaluated in PLFs and LAM/TSC cells, in PLFs grown for 48 hours in their own CM and in LAM/TSC cell CM. For the latter two groups, we reported the IL-8 secretion of PLFs after growing in CMs (their own and LAM/TSC cell CM) subtracted of the IL-8 levels of PLFs (PLFs in Figure 7c) and LAM/TSC cells (LAM/TSC cells in Figure 7c), in this way showing the IL-8 secretion of PLFs only when grown in CMs. We insert more detailed information in the legend of figure 7 and in the text (line 306-307) hoping that now the IL-8 data is more clearly presented.

Round 2
Reviewer 1 Report
The authors have nicely addressed all of my concerns. Very nice work